# The Role of Interleukin 6 (IL6), Cancer Antigen —125 (CA-125), and Human Epididymis Protein 4 (HE4) to predict tumor resectability in the advanced epithelial ovarian cancer patients

Syamel Muhammad[1]*, Reyhan Julio Azwan[1], Rauza Sukma Rita[2], Restu Susanti[3], Yusrawati[4]

1 Obstetrics and Gynecology Department, Medical Faculty of Andalas University, Padang, West Sumatera, Indonesia, 2 Biomedical Science Department, Medical Faculty of Andalas University, Padang, West Sumatera, Indonesia, 3 Nephrology Department, Medical Faculty of Andalas University, Padang, West Sumatera, Indonesia, 4 Fetomaternal Division, Obstetrics and Gynecology Department, Medical Faculty of Andalas University, Padang, West Sumatera, Indonesia

* syamelmuhammad@med.unand.ac.id

## Abstract

### Introduction

A study of tumor resectability in pre-operative patients with advanced epithelial ovarian cancer is required to predict primary surgical benefits accurately. This study aims to investigate IL6, CA-125 and HE4 to predict tumor resectability in the pre-operative patients with advanced epithelial ovarian cancer.

### Methods

This cross-sectional study was conducted in the polyclinic, oncology and gynecology inpatient room of Dr. M. Jamil Padang Hospital from June until December 2022. Advanced epithelial ovarian cancer stage based on histology result from FIGO stages IIIB–IVA. IL6, CA-125, and HE4 were measured using ECLIA (electrochemiluminescence immunoassay). Categorical data were assessed using Chi-square and Mann-Whitney tests. Numerical variable correlations were analyzed using Pearson Correlation tests. While the correlation between numerical and nominal variables was analyzed using the Eta correlation test. A p-value of <0,05 was considered a significant correlation. The cut-off value of serum IL6, CA-125, and HE4 was determined with a ROC curve. The sensitivity and specificity of each clinical parameter were calculated.

### Results

There was a significant difference in IL-6 (1328 vs 752 pg/ml; p<0,001), CA-125 (1260,5 vs 819,5 U/ml; p<0,001), and HE4 levels (1320 vs 760 pmol/L; p<0,001) between patients with

**Data Availability Statement:** Data cannot be shared publicly because of patients' confidentiality. Data are available from The Universitas Andalas Institutional Data Access / Ethics Committee (contact via ysyafrita@yahoo.com) for researchers who meet the criteria for access to confidential data.

**Funding:** This research received funding from Institute for Research and Community Services, Universitas Andalas (funding number: 38/UN.16.02/Fd/PT.01.03/2022). The funders had no role in study design, data collection and analysis, decision to publish, or preparation of the manuscript.

**Competing interests:** The authors have declared that no competing interests exist.

tumor resectability of > 1 cm (suboptimal) vs < 1 cm (optimal). There was a correlation between IL6 (r = 0,832), CA-125 (r = 0,716), and HE4 (r = 0,716) with tumor resectability.

## Conclusion

Measuring IL6, CA-125, and HE4 levels is useful for clinicians to predict tumor resectability in pre-operative patients with advanced epithelial ovarian cancer.

## Introduction

Epithelial ovarian cancer is a neoplasm that develops in the epithelial tissue covering the outside of an ovary. It is the most common type of ovarian cancer, contributing to 90% of ovarian cancer types [1]. Worldwide, ovarian cancer was the third most common gynecological cancer in 2020, with 313,959 new cases of ovarian cancer recorded globally [2]. In the United States, approximately 22,240 new cases of ovarian cancer were diagnosed and 14,070 ovarian cancer deaths [1]. In Asia/Pacific, the cases of ovarian cancer were 9.2/100,000, and in Indonesia, the incidence and mortality are the third-highest, accounting for 7% of malignancy in women, with 9581/14,896 deaths in 2020 [3].

Tumor markers, substances produced by cancer cells or normal cells in response to cancer growth, play a crucial role in diagnosing and managing epithelial ovarian cancer. Their advantages include early detection, monitoring treatment response, recurrence surveillance, non-invasive testing, and providing prognostic information. However, these markers also have limitations, such as lack of specificity, potential for false results, variability among individuals, limited effectiveness in early-stage detection, and cost and availability concerns. To maximize their benefits, tumor markers should be used in conjunction with other diagnostic methods, ensuring accurate and comprehensive care for ovarian cancer patients [4, 5].

A standard diagnostic assessment for a woman with suspected advanced ovarian cancer includes a physical examination, ultrasonography, serum cancer antigen (CA-125) measurement and imaging analysis (CT scan or MRI). To obtain an accurate prediction of primary surgical benefits, these standard procedures had low accuracy in predicting the outcomes of primary surgery with a deposition of tumor resectability > 1 cm. Furthermore, the location of the tumor affects the metastatic tumor resectability, and an optimum cytoreduction is challenging with a spreading disease into the diaphragm, liver parenchyma, small intestinal surface area, lesser omentum, or hepatic portal [6].

Cancer antigen 125 (CA-125) is a protein detected in the blood and commonly used to detect early ovarian cancer. The elevation of CA-125 is also associated with other malignancies, such as pancreatic, lung, breast, colorectal, and benign ovarian cysts. The evaluation of CA-125 has a low specificity to diagnose ovarian cancer but the potential to assess, monitor, and evaluate the responses of ovarian cancer therapy [7]. The production of IL6 is a result of proinflammatory cytokines, such as TNFα. In ovarian cancer, IL6 has a direct stimulus towards cancer cells via various mechanisms that contribute to the cell cycle and cancer cell growth [8]. Human epididymis protein 4 (HE4) is a whey acid protein precursor for epididymis secretory E4 with a molecular weight of 11 kDa. HE4 is expressed in various normal tissues, including respiratory and reproduction tissues. HE4 is overexpressed in ovarian epithelial cancer cells. Studies reported good performances of circulating HE4 for ovarian cancer detection and showed a promising role as a prognostic biomarker. In vitro and in vivo studies reported that HE4 plays a role in several molecular pathways associated with cell proliferation, tumor growth, and metastasis in ovarian cancer. Furthermore, the

combination of CA-125 and HE4 had a high accuracy in detecting ovarian cancer and disease progression [9].

The main aim of this study is to investigate CA-125 and HE4 levels as the ovarian cancer markers and IL6 as a pro-inflammatory factor that contributes to the cell cycle and growth of ovarian cancer cells to predict tumor resectability in pre-operative patients with advanced epithelial ovarian cancer.

## Materials and methods

This cross-sectional study was conducted in the polyclinic, oncology and gynecology inpatient room of Dr. M. Jamil Padang Hospital from June until December 2022. The inclusion criteria were patients admitted to polyclinic, oncology and gynecology inpatient room, Dr. M. Jamil Padang Hospital with the advanced stage of epithelial ovarian cancer based on pathological anatomy results and willing to participate in this study. The exclusion criteria were (1) patients admitted to polyclinic, oncology and gynecology inpatient room, Dr. M. Jamil Padang Hospital with the advanced stage of epithelial ovarian cancer, but lost to follow-up; (2) accompanied with other malignant; (3) received pre-operative chemotherapy or radiotherapy; (4) diagnosed with renal and heart disease, stroke, diabetes mellitus, hyperthyroidism, hypothyroidism, and other treated or undergoing therapy of metabolic diseases; (5) and patients with benign tumor based on pathological anatomy result.

This study involved patients' medical reports, and all medical matters relating to this research are confidential. The ethical implications of this study followed the provisions of the Declaration of Helsinki and were approved by the ethical committee of Dr. M. Jamil Padang Hospital (Approval number: 774/UN.162.KEP-FK/2021). The participants approved and signed the informed consent.

### Sample analysis

The advanced epithelial ovarian cancer stage was measured under a microscope based on histology results FIGO stages IIIB–IVA [10]. Post-operative tumor resectability of optimal resection $\leq 1$ cm and suboptimal resection $> 1$ cm [11]. IL6, CA-125, and HE4 were measured using ECLIA (electrochemiluminescence immunoassay).

### Statistical analysis

All data were collected and analyzed using a computer program SPSS (Statistical Package for the Social Sciences). In the descriptive analysis, categorical data were reported in frequency distribution and percentage. Numerical data of pre-operative IL6, CA-125, and HE4 were presented as mean and standard deviation. Univariate and bivariate analyses were performed. Saphiro Wilks test was used to test data normality. Categorical data were assessed using Chi-square and Mann-Whitney tests. Numerical variable correlations were analyzed using Pearson Correlation tests. While the correlation between numerical and nominal variables was analyzed using the Eta correlation test. A p-value of <0,05 was considered a significant correlation. The cut-off value of serum IL6, CA-125, and HE4 was determined with a ROC curve. Sensitivity was defined as the number of patients with suboptimal resections correctly identified divided by the total number of patients with suboptimal resections. Specificity was defined as the number of patients with optimal resection correctly identified divided by the total number of patients with optimal resection.

## Results

### Patients' characteristics

Table 1 shows the patients' characteristics. The participants were dominated by age 40–60 years old (57,8%) with a BMI of 18,5–25 kg/m2 (51,6%) and without a history of menopause (51,3%). According to the parity, most of the patients did not have a pregnancy history (37,5%).

### Lab profiles

Table 2 shows lab profiles of hemoglobin, erythrocyte, leukocyte, thrombocyte, APTT, albumin, protein, IL6, CA-125, and HE4.

### Tumor marker profiles

Table 3 shows the differences between IL-6, CA-125, and HE4 based on tumor resectability. There was a significant difference in IL-6 levels between patients with tumor resectability of suboptimal resection > 1 cm vs optimal resection < 1 cm (1328 vs 752 pg/ml; p<0,001). There was a significant difference in CA-125 levels between patients with tumor resectability of suboptimal resection > 1 cm vs optimal resection < 1 cm (1260,5 vs 819,5 U/ml; p<0,001). There was a significant difference in HE4 levels between patients with tumor resectability of suboptimal resection > 1 cm vs optimal resection < 1 cm (1320 vs 760 pmol/L; p<0,001).

The correlation between tumor markers (IL6, CA-125, HE4) with tumor resectability is shown in Table 4. There was a positive correlation between IL-6 (r = 0,832; p-value = 0,000),

**Table 1. Patients' characteristics.**

| Variable | n (64) | % |
|---|---|---|
| **Age (years)** | | |
| <19 | 1 | 1,6 |
| 19–39 | 14 | 21,9 |
| 40–64 | 37 | 57,8 |
| >64 | 12 | 18,8 |
| **BMI (kg/m$^2$)** | | |
| <17 | 1 | 1,6 |
| 17-<18.5 | 7 | 10,9 |
| 18.5–25 | 33 | 51,6 |
| >25–27 | 10 | 15,6 |
| >27 | 13 | 20,3 |
| **Parity** | | |
| 0 | 24 | 37,5 |
| <3 | 17 | 26,6 |
| 3–5 | 16 | 25 |
| >5 | 7 | 10,9 |
| **Menopause** | | |
| Yes | 30 | 46,9 |
| No | 34 | 53,1 |
| **Resectability** | | |
| <1 (optimal) | 38 | 59,4 |
| > 1 (suboptimal) | 26 | 40,6 |

**Table 2. Lab profiles.**

| Variable | Mean | SD | Median | Min-max | 95% CI | Normality |
|---|---|---|---|---|---|---|
| Hemoglobin (g/dL) | 11,25 | 1,70 | 11,2 | 7,20–15,10 | 10,82–11,67 | 0,20 |
| Erythrocyte ($10^{12}$/L) | 4,26 | 0,65 | 4,26 | 2,52–5,72 | 4,09–4,42 | 0,20 |
| Leukocyte ($10^9$/L) | 8,70 | 3,47 | 7,96 | 2,72–19,13 | 7,84–9,57 | 0,00 |
| Thrombocyte ($10^9$/L) | 375,031 | 120,13 | 362 | 135–875 | 345,03–405,04 | 0,06 |
| APTT (seconds) | 26,56 | 8,07 | 26,80 | 2,30–80,50 | 24,54–28,57 | 0,00 |
| Albumin (mg/dL) | 3,95 | 0,64 | 4,1 | 1,80–5,50 | 3,79–4,11 | 0,07 |
| Protein (g/L) | 7,30 | 1.30 | 7,50 | 0,40–9,60 | 6,98–7,63 | 0,00 |
| IL6 (pg/ml) | 158,41 | 99,44 | 99,44 | 55,05–444,56 | 133,57–183,24 | 0,00 |
| CA-125 (U/ml) | 16406,73 | 11412,21 | 12252,50 | 6148–60768 | 13556,05–19257,42 | 0,00 |
| HE4 (pmol/L) | 473,14 | 398,50 | 321,58 | 145,95–2016,58 | 373,59–572,68 | 0,00 |

CA-125 (r = 0,716; p-value = 0,000), or HE4 (r = 0,716; p-value = 0,000) with tumor resectability.

Table 5 shows the ROC analysis of IL-6, CA-125, HE4, ROMA, and ROMA + IL-6. Both markers of ROMA and ROMA + IL6 had lower sensitivity (61,50%) and specificity (60,50%) compared to IL-6 (92,3% and 92,15%), CA-125 (88,55% and 89,5%), and HE4 (96,2% and 97,4%).

### Tumor marker profiles and the outcomes

Fig 1 displays the ROC and area under a curve of 98,9% with a p-value of <0,001 to predict the surgical outcomes of the patients. The cut-off points of IL6 were 128,53 pg/ml with sensitivity and specificity of 92,3% and 92,1%, respectively.

Fig 2 displays the ROC and area under a curve of 92,1% with a p-value of <0,001 to predict the surgical outcomes of the patients. The cut-off points of CA-125 were 13249,50 U/ml with sensitivity and specificity of 88,5% and 89,5%, respectively.

Fig 3 displays the ROC and area under a curve of 98,1% with a p-value of <0,001 to predict the surgical outcomes of the patients. The cut-off points of HE4 were 410,40 pmol/L with sensitivity and specificity of 96,2% and 97,4%, respectively.

## Discussion

This study highlights the importance of evaluating tumor marker (IL6, CA-125, and HE4) levels to predict tumor resectability in pre-operative patients with advanced epithelial ovarian cancer. Our study reports that IL6, CA-125, and HE4 levels were higher in the group with tumor resectability of suboptimal > 1 cm. Similar studies have also reported combining tumor markers to predict surgical outcomes in patients with advanced epithelial ovarian cancer. A study by Kampan et al. reported that IL6, CA-125, and HE4 had a high accuracy in diagnosing

**Table 3. The profiles of IL-6, CA-125, and HE4 levels based on tumor resectability.**

| Variable | Resectability | | p-value |
|---|---|---|---|
| | <1 cm (optimal) | >1cm (suboptimal) | |
| IL6 (pg/ml) | 752 | 1328 | 0,000 |
| CA-125 (U/ml) | 819,5 | 1260,5 | 0,000 |
| HE4 (pmol/L) | 760 | 1320 | 0,000 |

Table 4. Correlation between IL-6, CA-125, or HE4 with tumor resectability.

| Variable | Coefficient correlation R | p-value |
|---|---|---|
| IL6 (pg/ml) | 0,832 | 0,000 |
| CA-125 (U/ml) | 0,716 | 0,000 |
| HE4 (pmol/L) | 0,716 | 0,000 |

or predicting the prognosis of optimum cytoreduction in advanced epithelial ovarian cancer with sensitivity and specificity of 100% [8]. Evaluation of IL6 and CA-125 has also been reported to increase the prediction accuracy and determine better management for ovarian cancer [12]. A similar study by Kurniadi et al. found a correlation between pre-operative IL6 and CA-125 levels to predict tumor outcomes [13]. A combination between CA-125 and HE4 has also been reported in a study by Feng et al., who found that the levels of pre-operative CA-125 $\geq$ 313,6 U/mL and HE4 $\geq$ 777,1 pmol/L increased the occurrence of suboptimum cytoreduction in the advanced ovarian cancer [14].

According to the Eta correlation test, the difference levels of IL6, CA-125, and HE4 between groups with tumor resectability of suboptimal > 1 cm and optimal < 1 cm also showed a strong positive correlation to the tumor resectability. Additionally, the high sensitivity and specificity of IL6, CA-125, and HE4 showed that evaluation of their levels could be utilized to predict surgical outcomes. Previous studies have also reported high sensitivity and specificity of using a single tumor marker IL6, CA-125, HE4, or combinations to predict ovarian cancer surgical outcomes [8, 13]. The cut-off point value of IL6, CA-125, and CA-125 in this study was also higher than in previous studies. The number of subjects in our study was higher than the other studies may explain the reason for higher sensitivity and sensitivity, as well as the result of the cut-off point value obtained.

The role of IL6 in promoting cellular proliferation in various types of cancer, such as breast, lung, and colorectal cancer, has been well reported. Therefore, the higher IL6 serum levels may also explained the association of pro-inflammatory mediator in the development of advanced ovarian epithelial cancer [15]. The production of cytokines during the inflammation process contributes to the binding and activation of toll-like receptors (TLRs) on the cellular surface. This mechanism involves mitogen-activated protein kinases (MAPKs) p38 and C-jun N-terminal Kineas (JNK) and the transcription factors NF-kB. The MAPK pathway will regulate cellular proliferation, differentiation, growth, migration, and apoptosis by upregulating AP-1, c-Jun, and FOS transcription factors and activation of NF-kB and STATS. Furthermore, NF-kB NF-kB and AP-1 will regulate the production of pro-inflammatory IL6 [16].

Inteleukin-6 is a pro-inflammatory cytokine located in the tumor microenvironment. The production results from local pro-inflammatory cytokine responses, such as TNF-α, expressed mainly in ovarian cancer. IL6 recruits the neutrophils and plays a role in cellular migration, growth, activation, and T- and B-lymphocyte differentiation into plasma cells [17].

Table 5. ROC analysis using IL-6, CA-125, HE4, ROMA, and ROMA + IL-6.

| Marker | AUC | P-value | 95%CI | Sensitivity | Specificity | COP |
|---|---|---|---|---|---|---|
| IL-6 | 0,989 | 0,00 | 0,971–1 | 92,3% | 92,15% | 128,53 pg/ml |
| CA-125 | 0,921 | 0,00 | 0,842–0,999 | 88,55% | 89,5% | 13249,50 u/ML |
| HE4 | 98,1 | 0,00 | 0,946–1 | 96,2% | 97,4% | 410,49 pmol/L |
| ROMA | 0,77 | 0,00 | 0,647–0,889 | 61,50% | 60,50% | 6075.7450 |
| ROMA + IL6 | 0,77 | 0,00 | 0,647–0,889 | 61,50% | 60,50% | 6267.8800 |

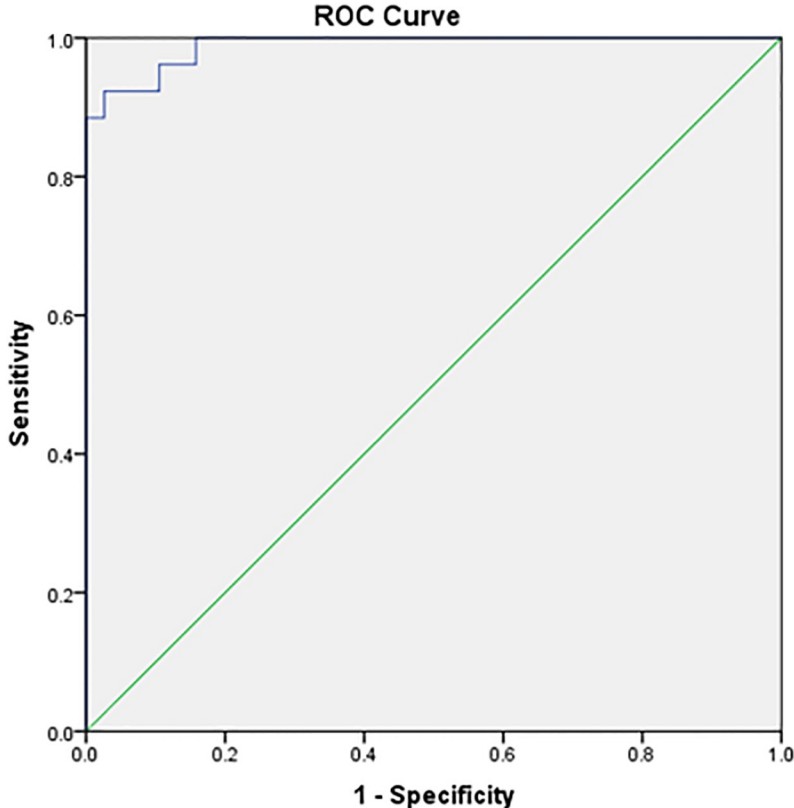

**Fig 1. Surgical outcomes of IL6 levels.**

Physiologically, IL6 plays an important role in ovarian follicle development in the angiogenesis process by inducing the phosphorylation of STAT3 and MAPK within endothelial cells to enhance cellular migration [18]. Ovarian cancer cells secrete IL6 and IL6Rα (soluble form sIL-6R) through trans-signaling sIL6R-IL6 in the pro-inflammatory microenvironment cells that do not express transmembrane receptor to facilitate angiogenesis via increasing the VEGF expressions [19]. The rapid responses of IL6 to internal and external stimuli make it sensitive to be utilized in the clinical setting [15]. Elevation of IL6 levels may result from ovarian cancer cell proliferation via the activation MAPK-ERK-Akt (protein kinase B) signaling pathway. In the ascites, activation of ERK showed an increase in tumor cellular migration. Additionally, the production of IL6 by M2 macrophage can induce cancer cellular proliferation via STAT3 activation in ascites patients with advanced ovarian cancer [20]. Furthermore, producing MMPs in ovarian cancer cells by IL6 can enhance invasive and tumorigenesis performance [17] The increase of IL6 levels in peritoneum fluid has also shown that the role of JAK-STAT helps the invasive process of ovarian cancer [21].

Although we found the sensitivity and specificity were high in measuring CA-125 to predict surgical outcomes of cytoreduction, some studies also showed that measuring CA-125 was not accurate in predicting optimum cytoreduction surgical outcomes, such as a study by Angioli et al. reported the cut-off point of CA-125 was 414 IU/mL with 58,3% and 84% sensitivity and specificity, respectively [22]. However, a meta-analysis by Kang et al. found that although the levels of CA-125 were not significant to predict optimum cytoreduction, pre-operative CA-125 ≥ 500 IU/mL was considered as the risk factor of suboptimum cytoreduction with 50% of

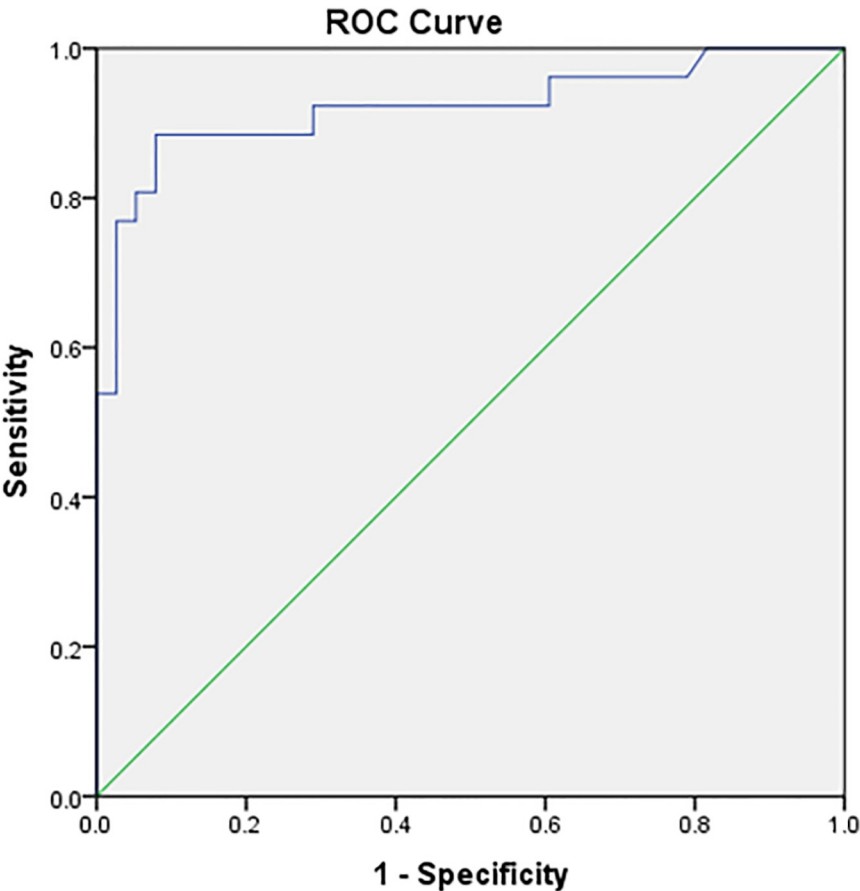

**Fig 2. Surgical outcomes of CA-125 levels.**

cases required more extensive abdominal surgical procedure. Therefore, a higher pre-operative CA-125 represents the difficulty and complexity of surgical procedures that need to be considered [23].

The increase of CA-125 levels was also reported in other malignancies, such as pancreatic, lung, breast, and colorectal cancer, as well as in ovarian cysts. It was reported that patients with a consistently high CA-125 show the possibility of tumor resectability [17]. An in vitro study found that CA-125 was a potent NK cell inhibitor that reduces CD16 expression on the NK cells (FcγRIII) surface. The binding of CA-125 into the NK cells via galectin-1 facilitated the attachment into the NK cells, promoted metastases, and prevented immune responses [8]. CA-125 also binds with mesothelin, a protein expressed by ovarian, lung, and pancreas cancers, in the normal mesothelium. The mesothelin and protein CA-125 interaction plays an important role in the implantation of ovarian cancer cells into the peritoneum wall [24, 25].

The role of CA-125 in predicting suboptimum cytoreduction has been investigated in a multicenter prospective study involving patients with stages III and IV who underwent primary cytoreduction surgery. Six criteria were classified that associated with suboptimum cytoreduction (tumor resectability > 1 cm), including the levels of CA-125 > 500 IU/mL [26]. Additionally, the use of CA-125 is also potentiated to predict recurrent ovarian cancer.

Early screening is a prevention strategy to reduce the mortality of ovarian cancer. A high expression of CA-125 is often detected earlier than other clinical signs of physical examination

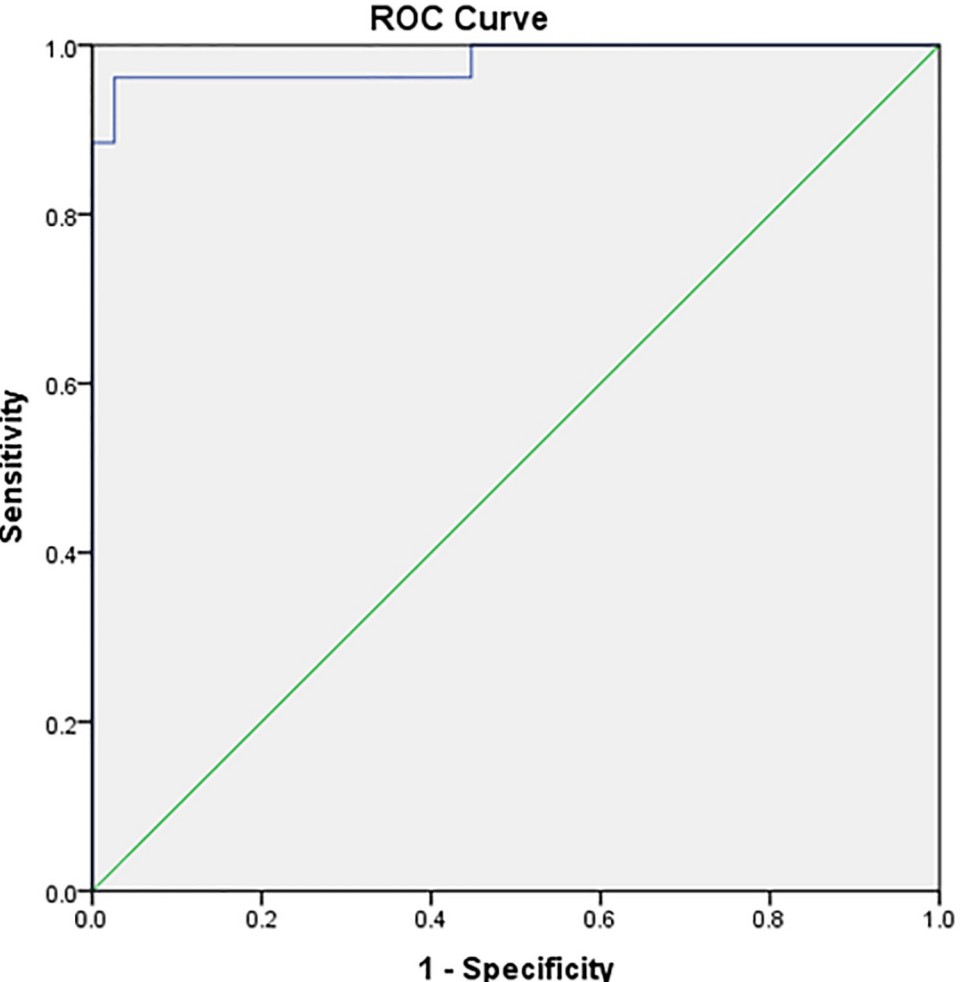

**Fig 3. Surgical outcomes of HE4 levels.**

and imaging in 80% of ovarian cancer. Currently, the evaluation of CA-125 is involved in the monitoring standard procedures of the patients who underwent remission to detect possible recurrent cancer and to anticipate better treatment strategy [27].

The role of HE4 in predicting the surgical outcomes of ovarian cancer with high sensitivity and specificity has also been reported in previous studies. A study using the combination of tumor markers by Holcomb et al. showed that HE4 was more accurate than other tumor markers, such as CA-125, in predicting epithelial ovarian cancer (88,9% sensitivity and 91,8% specificity) [28]. Similar results have also been reported by Plotti et al. with 75,53% sensitivity and 100% specificity [29]. Additionally, the role of HE4 in monitoring ovarian cancer was more accurate than the benign (98,6% sensitivity and 73,3% specificity) [30].

Previous studies reported the necessity to evaluate early HE4 levels to predict optimum cytoreduction of advanced epithelial ovarian cancer. A prospective study by Angioli et al. in 57 advanced ovarian epithelial cancer showed that HE < 262 pmol/L had 86,1% sensitivity and 89,5% specificity with positive and negative predictive values of 93,9% and 77%, respectively, to predict the success of optimum cytoreductive surgery. Furthermore, HE4 < 262 pmol/L and ascites volume < 500mL had 100% sensitivity and 89,5% specificity with positive and

negative predictive values of 94% and 100%, respectively [22]. A similar study was also reported by Shen et al., which also showed that the detection of HE4 was even better than CA-125 in predicting the resectability of ovarian cancer [31]. A Study by Moore et al. reported that HE4 levels could be utilized as a single tumor marker with sensitivity and specificity of 72,9% and 95%, respectively [32]. Furthermore, other studies also reported the same results showing that single HE4 levels detection was more accurate than the combination with other tumor markers (especially CA-125) [28, 30].

The human epididymis protein 4 (HE4) gene is expressed with an exceeded amount in ovarian cancer. The elevation of HE4 was reported in almost all ovarian cancer patients (31/32 patients with > 150 pmol/L) [33]. Therefore, HE4 is a promising tumor marker to diagnose, evaluate the prognosis, and follow up the ovarian cancer patients. The immunoreactivity of HE4 showed the highest levels in various normal human tissues, including the epithelial glands of the genital tract [34]. Various in vitro and in vivo studies have been conducted to investigate the role of HE4 in the proliferation and development of ovarian cancer cells. Wang et al. reported that the given cells with HE4 recombinant group showed significant statistical results regarding the number of colonization among the control group. Additionally, the ablation of HE4 via shRNA will result in the reduction of cellular proliferation. An in vitro study showed the role of HE4 in cellular proliferation by regulating the cell cycle, particularly the transition of G0/G1. The loss of HE4 will terminate G0/G1 cell cycle and the G1 to S transition phase. Conversely, the stimulus of HE4 recombinant will enhance the number of cells in the G2/M phase [35].

In the management to predict tumor resectability, the evaluation of tumor marker levels (IL-6, CA-125, and HE4) is paramount importance, particularly to achieve sub-optimal debulking surgery [36]. However, in patients with low probability of achieving optimal debulking surgery, neoadjuvant chemotherapy followed by internal debulking surgery increases the chance of optimal surgery [37]. As debulking surgery is the primary outcome of most patients with advanced ovarian cancer, different approaches have been developed to improve outcomes for ovarian cancer patients, including the incorporation of targeted therapy, such as anti-VEGF agents, immunotherapy, and PARP inhibitors [38]. For instance, PARP inhibitors have emerged as a significant breakthrough in both newly diagnosed and recurrent cases, especially in patients with BRCA mutations [39]. Furthermore, the success of immunotherapy in various cancer types has sparked interest in its potential application in platinum-resistant ovarian cancer [40]. These novel therapies offer hope for patients facing limited treatment options and hold the potential to improve long-term outcomes.

The limitation in this study that we conducted in a single center with 63 patients involved. A multicenter study with larger samples is required for the future studies.

## Conclusion

Measuring IL6 (cut-off points = 128,53 pg/ml; 92,3% sensitivity and 92,1% specificity), CA-125 (cut-off points = 13249,50 U/ml; 88,5% sensitivity and 89,5% specificity), and HE4 (cut-off points = 410,40 pmol/L; 96,2% sensitivity and 97,4% specificity) levels is useful for clinician to predict tumor resectability in the preoperative patients with advanced epithelial ovarian cancer.

## Supporting information

**S1 Checklist. STROBE statement—Checklist of items that should be included in reports of observational studies.**
(DOCX)

**S1 File.**
(PDF)

**S2 File.**
(PDF)

## Acknowledgments

We are grateful to all staff at Hospital M. Djamil Padang who facilitated us in data collection, and to all participants who had been willing to participate in this study.

## Author Contributions

**Conceptualization:** Syamel Muhammad, Rauza Sukma Rita.

**Data curation:** Rauza Sukma Rita, Yusrawati.

**Funding acquisition:** Syamel Muhammad.

**Investigation:** Syamel Muhammad, Rauza Sukma Rita, Yusrawati.

**Methodology:** Syamel Muhammad.

**Project administration:** Syamel Muhammad.

**Resources:** Rauza Sukma Rita, Restu Susanti.

**Supervision:** Yusrawati.

**Validation:** Restu Susanti.

**Visualization:** Reyhan Julio Azwan, Rauza Sukma Rita.

**Writing – original draft:** Syamel Muhammad, Yusrawati.

**Writing – review & editing:** Rauza Sukma Rita, Yusrawati.

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
