## [Decision Letter · Decision Letter 0]

6 Jul 2023

PONE-D-23-10846The Role of Interleukin 6 (IL6), Cancer Antigen – 125 (CA-125), and Human Epididymis Protein 4 (HE4) to Predict Tumor Resectability in the Advanced Epithelial Ovarian Cancer PatientsPLOS ONE

Dear Dr. Muhammad,

Thank you for submitting your manuscript to PLOS ONE. After careful consideration, we feel that it has merit but does not fully meet PLOS ONE’s publication criteria as it currently stands. Therefore, we invite you to submit a revised version of the manuscript that addresses the points raised during the review process.

We look forward to receiving your revised manuscript.

Kind regards,

Andrea Giannini

Academic Editor

PLOS ONE

Journal Requirements:

"This research received funding from Institute for Research and Community Services, Universitas Andalas (funding number: 38/UN.16.02/Fd/PT.01.03/2022)."

"The authors have no conflicts of interest to declare. All co-authors have seen and agree with the contents of the manuscript and

there is no financial interest to report. We certify that the submission is original work and is not under review at any other publication"

**Additional Editor Comments:**

Dear authors,

the topic of the present article titled “The Role of Interleukin 6 (IL6), Cancer Antigen – 125 (CA-125), and Human Epididymis Protein 4 (HE4) to Predict Tumor Resectability in the Advanced Epithelial Ovarian Cancer Patients” is very interesting, the paper and the aim falls within the scope of the journal but the article needs major improvements.

The introduction, material and method section and tables should be modified and improved.

The manuscript should be organized better and English should be improved.

I suggest improving the manuscript with the reviewers' comments.

Reviewers' comments:

Reviewer's Responses to Questions

**Comments to the Author**

1. Is the manuscript technically sound, and do the data support the conclusions?

Reviewer #1: Partly

Reviewer #2: No

Reviewer #3: Yes

2. Has the statistical analysis been performed appropriately and rigorously? 

Reviewer #1: No

Reviewer #2: No

Reviewer #3: Yes

3. Have the authors made all data underlying the findings in their manuscript fully available?

Reviewer #1: Yes

Reviewer #2: No

Reviewer #3: Yes

4. Is the manuscript presented in an intelligible fashion and written in standard English?

Reviewer #1: No

Reviewer #2: No

Reviewer #3: Yes

5. Review Comments to the Author

Reviewer #1: I would like the authors to emphasise a little more the advantages and disadvantages of tumour markers.

Predicting tumor resectability in ovarian cancer typically relies on imaging studies such as computed tomography (CT), magnetic resonance imaging (MRI), and surgical exploration. These methods allow clinicians to visualize the tumor size, location, and involvement with nearby structures, helping determine if complete surgical resection is feasible. Biomarkers like IL-6, CA 125, and HE4 can provide valuable diagnostic and prognostic information, but they are not specific indicators of tumor resectability.

It's worth noting that ongoing research aims to identify novel biomarkers and develop more accurate predictive models for tumor resectability in ovarian cancer. However, the current understanding suggests that IL-6, CA 125, and HE4 are not directly predictive of tumor resectability in ovarian cancer

The Risk of Ovarian Malignancy Algorithm (ROMA) is a diagnostic tool that combines the levels of CA 125 and HE4 along with menopausal status to provide an improved prediction of ovarian cancer risk.

The ROMA index has been found to be useful in distinguishing between benign and malignant ovarian tumors, as well as in identifying patients at high risk for ovarian cancer. When used in combination with imaging studies, it can help guide treatment decisions and improve patient management.

However, the ROMA index may indirectly assist in predicting tumor resectability by providing additional information about the likelihood of malignancy. A higher ROMA index indicates a higher risk of ovarian cancer, which may suggest a more aggressive tumor requiring comprehensive surgical resection. Conversely, a lower ROMA index might suggest a lower risk of malignancy and potentially a better chance of complete tumor resection.

It's important to note that the ROMA index should be interpreted in conjunction with other clinical and imaging findings to make informed decisions about tumor resectability. The expertise and judgment of healthcare professionals experienced in managing ovarian cancer play a crucial role in assessing tumor resectability and determining the appropriate treatment approach.

I suggest that you calculate the ROMA index and include these results and try to plug IL-6 into this algorithm.

Furthermore, in some parts the text is difficult to understand. The part where you describe the results in the text in relation to Table 2 is very confusing. You can just mention the results of the other parameters, or you can just choose the text or just the table.

Reviewer #2: The second paragraph of the introduction is on the subject of early detection of ovarian cancer and is a complete other subject than this study. Please remove " Since the detection were commonly in the advanced stage, ovarian cancer is considered as a health global concern. A total of 75% ovarian cancer patients were detected in the advanced stage after

metastasis. In the early stage, ovarian cancer patients are asymptomatic and preferred to visit when

the cancer has been progressed into the advanced stage. The detection of advanced stage

contributed to the less than 30% of 5-year survival rates of the patients, conversely, the early

identification will improve 5-year survival rates up to 90%. Therefore, it is essential to perform a

study investigating the screening method to diagnose asymptomatic patients with early-stage

ovarian cancer to reduce the ovarian cancer mortality rates."

In the third paragraph you describe optimum cytoreduction (tumor reduction until <1cm). In more recent publications the importance of complete cytoreduction (until no visible tumour) is indicated as the preferrable outcome of cytoreduction since survival is better in this group compared to optimal cytoreduction. It is preferrable to have complete cytoreduction as primary outcome of resectability and not optimal cytordeuction.

Statistical analysis: you write 'All data were collected and analyzed using a computer program.'. What computerv programm?

Results "According to the parity, most of the patients were not having history of pregnancy (37,5 %)" This is strange: most had a parity, 37.5% not.

" The PCI score of the patients were dominated with score of 0 (50%)" This means that 50% had no visible diease!!

In results: as discussed before: complete resection is more important for survival than optimal resection: results should be re-analyzed for complete resection. And....what with the 50 that had a PCI score of 50??

Table 4, 5 and 6 should be joint together in 1 table

I did not review the discussion since I think the problems with PCI=0 and the complete vs optimal cytoreduction should be cleared first.

Reviewer #3: I read with great interest the Manuscript titled " The Role of Interleukin 6 (IL6), Cancer Antigen – 125 (CA-125), and Human Epididymis Protein 4 (HE4) to Predict Tumor Resectability in the Advanced Epithelial Ovarian Cancer Patients" which falls within the aim of the Journal. In my opinion, this topic analyzed is interesting enough to attract readers’ attention.

Although the manuscript can be considered already of good quality, I would suggest the following recommendations:

- I suggest round of language revision, in order to correct few typos and improve readability;

- Discussions can be expanded and improved by citing relevant articles about use of innovative therapeutic strategies in recurrent and advanced ovarian cancer. I would be glad if the authors discuss this important point, referring to PMID: 37314974 and 32518015

Because of these reasons, the article should be revised and completed. Considered all this points, I think it could be of interest for the readers and, in my opinion, it deserves the priority to be published after minor revisions.

6. PLOS authors have the option to publish the peer review history of their article (what does this mean?). If published, this will include your full peer review and any attached files.

Reviewer #1: No

Reviewer #2: No

Reviewer #3: **Yes: **Giorgio Bogani

---

## [Author Response · Author response to Decision Letter 0]

30 Aug 2023

EDITORS COMMENTS:

1. The introduction, material and method section and tables should be modified and improved. 

 • The authors have improved based on the reviewers’ comments

2. The manuscript should be organized better and English should be improved.

 • We have organized and improved the English

3. I suggest improving the manuscript with the reviewers' comments.

 • Done

REVIEWERS COMMENTS:

REVIEWER 1

1. I would like the authors to emphasise a little more the advantages and disadvantages of tumour markers.

 • We have emphasized the advantages and disadvantages 

2. Predicting tumor resectability in ovarian cancer typically relies on imaging studies such as computed tomography (CT), magnetic resonance imaging (MRI), and surgical exploration. These methods allow clinicians to visualize the tumor size, location, and involvement with nearby structures, helping determine if complete surgical resection is feasible. Biomarkers like IL-6, CA 125, and HE4 can provide valuable diagnostic and prognostic information, but they are not specific indicators of tumor resectability.

 • Yes. The role of imaging studies and surgical exploration are important in determining if complete surgical resection is feasible. However, based on our study, it shows that the biomarker IL-6, CA 125, and HE4 is also useful to predict surgical tumor resection. Therefore, a combination between them may provide higher accuracy.

3. It's worth noting that ongoing research aims to identify novel biomarkers and develop more accurate predictive models for tumor resectability in ovarian cancer. However, the current understanding suggests that IL-6, CA 125, and HE4 are not directly predictive of tumor resectability in ovarian cancer

 • Yes, it is not directly.

4. The Risk of Ovarian Malignancy Algorithm (ROMA) is a diagnostic tool that combines the levels of CA 125 and HE4 along with menopausal status to provide an improved prediction of ovarian cancer risk. The ROMA index has been found to be useful in distinguishing between benign and malignant ovarian tumors, as well as in identifying patients at high risk for ovarian cancer. When used in combination with imaging studies, it can help guide treatment decisions and improve patient management. However, the ROMA index may indirectly assist in predicting tumor resectability by providing additional information about the likelihood of malignancy. A higher ROMA index indicates a higher risk of ovarian cancer, which may suggest a more aggressive tumor requiring comprehensive surgical resection. Conversely, a lower ROMA index might suggest a lower risk of malignancy and potentially a better chance of complete tumor resection. It's important to note that the ROMA index should be interpreted in conjunction with other clinical and imaging findings to make informed decisions about tumor resectability. The expertise and judgment of healthcare professionals experienced in managing ovarian cancer play a crucial role in assessing tumor resectability and determining the appropriate treatment approach. I suggest that you calculate the ROMA index and include these results and try to plug IL-6 into this algorithm.

 • We have calculated the ROMA index and the plug it with IL-6, as displayed in table 5.

5. Furthermore, in some parts the text is difficult to understand. The part where you describe the results in the text in relation to Table 2 is very confusing. You can just mention the results of the other parameters, or you can just choose the text or just the table.

 • We have revised the explanation of table 2.

REVIEWER 2

1. The second paragraph of the introduction is on the subject of early detection of ovarian cancer and is a complete other subject than this study. Please remove " Since the detection were commonly in the advanced stage, ovarian cancer is considered as a health global concern. A total of 75% ovarian cancer patients were detected in the advanced stage after metastasis. In the early stage, ovarian cancer patients are asymptomatic and preferred to visit when the cancer has been progressed into the advanced stage. The detection of advanced stage contributed to the less than 30% of 5-year survival rates of the patients, conversely, the early identification will improve 5-year survival rates up to 90%. Therefore, it is essential to perform a study investigating the screening method to diagnose asymptomatic patients with early-stage ovarian cancer to reduce the ovarian cancer mortality rates."

 • Removed.

2. In the third paragraph you describe optimum cytoreduction (tumor reduction until <1cm). In more recent publications the importance of complete cytoreduction (until no visible tumour) is indicated as the preferrable outcome of cytoreduction since survival is better in this group compared to optimal cytoreduction. It is preferrable to have complete cytoreduction as primary outcome of resectability and not optimal cytordeuction.

 • We prefer to choose optimal because we want to assess the diameter of tumor residue in < 2 cm, and not complete cytoreduction since it does not show any residue microscopically. 

3. Statistical analysis: you write 'All data were collected and analyzed using a computer program.'. What computer program?

 • Already explained.

4. Results "According to the parity, most of the patients were not having history of pregnancy (37,5 %)" This is strange: most had a parity, 37.5% not. " The PCI score of the patients were dominated with score of 0 (50%)" This means that 50% had no visible disease!!. In results: as discussed before: complete resection is more important for survival than optimal resection: results should be re-analyzed for complete resection. And....what with the 50 that had a PCI score of 50??

 • We prefer to choose optimal and not complete resection since we want to measure the diameter of residue. 

 • We have decided to remove the explanation of PCI score, since it does not associate with the measurement of the biomarkers.

5. Table 4, 5 and 6 should be joint together in 1 table

 • We have joined them.

6. I did not review the discussion since I think the problems with PCI=0 and the complete vs optimal cytoreduction should be cleared first.

 • We have explained the reason of choosing optimal cytoreduction.

REVIEWER 3

1. I suggest round of language revision, in order to correct few typos and improve readability;

 • Already revised.

2. Discussions can be expanded and improved by citing relevant articles about use of innovative therapeutic strategies in recurrent and advanced ovarian cancer. I would be glad if the authors discuss this important point, referring to PMID: 37314974 and 32518015

 • We have added the current information regarding the innovative therapeutic strategies based on the reference suggested in the discussion section.

3. Because of these reasons, the article should be revised and completed. Considered all this points, I think it could be of interest for the readers and, in my opinion, it deserves the priority to be published after minor revisions.

 • We have revised and completed based on editors’ and reviewers’ comments

---

## [Decision Letter · Decision Letter 1]

18 Sep 2023

The Role of Interleukin 6 (IL6), Cancer Antigen – 125 (CA-125), and Human Epididymis Protein 4 (HE4) to Predict Tumor Resectability in the Advanced Epithelial Ovarian Cancer Patients

PONE-D-23-10846R1

Dear Dr. Muhammad,

We’re pleased to inform you that your manuscript has been judged scientifically suitable for publication and will be formally accepted for publication once it meets all outstanding technical requirements.

Kind regards,

Andrea Giannini

Academic Editor

PLOS ONE

Additional Editor Comments (optional):

The manuscript has been modified with the comments of the reviewers. It is now ready to be published.

Reviewers' comments:

Reviewer's Responses to Questions

**Comments to the Author**

1. If the authors have adequately addressed your comments raised in a previous round of review and you feel that this manuscript is now acceptable for publication, you may indicate that here to bypass the “Comments to the Author” section, enter your conflict of interest statement in the “Confidential to Editor” section, and submit your "Accept" recommendation.

Reviewer #3: All comments have been addressed

2. Is the manuscript technically sound, and do the data support the conclusions?

Reviewer #3: Yes

3. Has the statistical analysis been performed appropriately and rigorously? 

Reviewer #3: Yes

4. Have the authors made all data underlying the findings in their manuscript fully available?

Reviewer #3: Yes

5. Is the manuscript presented in an intelligible fashion and written in standard English?

Reviewer #3: Yes

6. Review Comments to the Author

Reviewer #3: The quality of the manuscript has improved thanks to the changes made. I think it could be of interest to the readers and, in my opinion, it deserves the priority to be published.

7. PLOS authors have the option to publish the peer review history of their article (what does this mean?). If published, this will include your full peer review and any attached files.

Reviewer #3: **Yes: **Giorgio Bogani

---

## [Editor Report · Acceptance letter]

25 Sep 2023

PONE-D-23-10846R1 

The Role of Interleukin 6 (IL6), Cancer Antigen – 125 (CA-125), and Human Epididymis Protein 4 (HE4) to Predict Tumor Resectability in the Advanced Epithelial Ovarian Cancer Patients 

Dear Dr. Muhammad:

I'm pleased to inform you that your manuscript has been deemed suitable for publication in PLOS ONE. Congratulations! Your manuscript is now with our production department. 

Kind regards, 

on behalf of

Dr. Andrea Giannini 

Academic Editor

PLOS ONE